# The Role of Trust and Risk in Citizens' E-Government Services Adoption: A Perspective of the Extended UTAUT Model

Wenjuan Li [1,2,3]

1 School of Public Management and Public Policy, Tsinghua University, Beijing 100084, China; liwj0915@126.com
2 Centre for Crisis Management Research, Tsinghua University, Beijing 100084, China
3 Centre for Social Risk Assessment in China, Tsinghua University, Beijing 100084, China

**Abstract:** This research particularly aims to investigate how trust and perceived risk influence citizens' e-government adoption. The findings of the study reveal that citizens' trust of the government (TOG) and trust of the internet (TOI) positively affect citizens' e-government adoption (EGA); perceived risk (PR) is negatively associated with citizens' EGA. Interestingly, this study also demonstrates the negative moderating effect of PR on the relationship between TOG and EGA, TOI and EGA. The results also indicate that performance expectancy (PE), effort expectancy (EE), social influence (SI), and facilitation conditions (FC) influence citizens' EGA positively. Lastly, implications for practice and research are discussed.

**Keywords:** trust; perceived risk; e-government adoption; e-government services





## 1. Introduction

E-Government in China has achieved remarkable achievements in the context of digital transformation over the last few years, and can provide a variety of online services to citizens. Despite the governments' growing provision of online services, citizens are still more likely to use traditional channels (e.g., 12,345 hotline or in-person visits to the Government Service Halls) than electronic channels to interact with the government. Many citizen users are reluctant to accept e-government services due to a lack of trust in the internet and e-government services platforms (websites, WeChat, Weibo, etc.), and the risk of exposure of personal data submitted electronically. These concerns are not without merit. Unlike offline services, e-government services are unique due to the distance and impersonal nature of the internet [1]. To make matters worse, (1) the application of emerging technologies, such as artificial intelligence, blockchain, big data, and cloud computing have objectively brought security and risk issues; (2) the cybersecurity awareness of Chinese netizens has begun to raise, and they have gradually paid more attention to their personal information and privacy security issues; (3) the global and Chinese internet security environment has become increasingly complex, and risk has gradually increased in recent years [2,3]. In this context, research on the influence of trust and the risk perception on citizens' behaviour can find some possible solutions to this problem.

The role of trust and perceived risk have been explored in numerous studies. These studies have already added trust or risk into information and technology acceptance models such as the technology acceptant model (TAM), theory of reasoned action (TRA), and the theory of planned behaviour (TPB), etc., to explain the direct effects of trust and the risk on the citizens' e-government adoption [4–7]. Several studies incorporated both trust and risk into the above models and tried to explore the influence and interactive impact of citizens' trust and risk on e-government adoption. Belanger and Carter developed a model composed of four fundamental constructs that impact citizens' intention to use e-government services: trust of the internet, trust of the government, disposition to trust,

and perceived risk [1]. Lee and Song (2013) extended the UTAUT model by adding organizational trust and perceived risk [8]. Carter et al. (2016) proposed a model that used the theory of reasoned action (TRA) as the conceptual foundation and added TOI, TOG, and risk perception [9]. Although these studies revealed the influence mechanism of trust and risk perception on user behaviour, they have some shortcomings. First, these studies rarely added government trust, technology trust, and risk perception into the research model at the same time, to investigate the influence mechanism of TOG, TOI, and risk perception on the core constructs of the information technology acceptance model and users' attention to use e-government. Second, these studies only explored the mediating effect of perceived risk between trust and citizens' e-government adoption, which rarely paid attention to the moderating effect of risk perception on the relationship between trust and e-government adoption. In addition, the samples of these studies were mostly from Europe, America, or South Korea, and there is a lack of empirical research from China. Simultaneously, few studies in China focused on the influence mechanism of trust and risk perception on user behaviour. The most important is that the conclusions of these studies were different, which means that the influence path of trust and risk on user's e-government adoption and the relationship between trust and risk are complicated; it requires further empirical studies to test.

To make up for these research gaps, the purpose of this study is to investigate how trust and perceived risk impact citizens' e-government adoption, as well as to examine the moderating effect of perceived risk between trust and e-government adoption, taking the Chinese e-government services as the application domain. The paper is structured as follows: Section 2 presents the research context, research hypotheses, and conceptual framework. Section 3 introduces the survey design and research methodology, followed by Section 4, which highlights the data analysis and results. Section 5 provides further discussion and conclusions, including theoretical and practical implications of this study, the limitations, and future studies.

## 2. Research Context, Hypotheses, and Methods

### 2.1. Research Contex

The Chinese government has attached great importance to the construction of e-government in recent years. The State Council of China issued "Notification of action plan for promoting The Development of Big Data" in 2015, and "Guidelines for actively promoting the "Internet +" initiative" in 2016, which point the way and provide top-level design for the development of e-government. After several years of construction, e-government in China has made remarkable achievements, which has played a major role in improving public services, strengthening social management, strengthening comprehensive supervision, improving macro-regulation, and promoting ecological protection, and became an indispensable tool to promote the modernisation of government governance systems and governance capabilities [10]. More and more online platforms are used to provide e-government services, including government websites, Weibo, WeChat, Tik Tok, Ali-pay, etc. Despite there being many online platforms that provide information and online transactions to citizens, one of the most comprehensive platforms is the government websites, which is the main channel for citizens to obtain information about laws and policies, reports, mayors' profiles, government departments, and administrative districts, as well as complete online transactions, such as driver's and marriage licenses, medical insurance, and infant registration, and participate in public affairs such as share opinions through questionnaires, etc. Therefore, according to the practice of e-government development in China, this study focuses on citizens' adoption of e-government websites.

### 2.2. Research Hypotheses

#### 2.2.1. Trust, Risk, and Citizens' E-Government Adoption

Trust, as a relationship between trustor and trustee, has attracted the attention of scholars in sociology, political science, and economics for nearly half a century. Their

studies focused on how trust affects organisations, society, and political systems. With the rapid development of e-government worldwide, more and more scholars have begun to pay attention to the impact of trust on e-government adoption. There are a variety of definitions and operational scales of trust in existing studies. Nevertheless, all these conceptions of trust are related to the trustee, namely the object of trust. For instance, Papadopoulos et al. (2010) proposed a comprehensive typology of trust in e-government (seven different types, conceptualised around the different targets they are related to), including trust in data, trust in service, trust in information, trust in the transaction, trust in government organisation, and institution-based trust [11]. In the case of e-government in China, the main objects of trust include public organisations which provide information and services for users, and the internet which delivers information and services to users. Furthermore, numerous studies have suggested that trust in e-government is composed of trust in an entity (trust in government) and trust in the reliability of the enabling technology [9]. Hence, this study focuses on both trusts: trust of the government (TOG) and trust of the internet (TOI).

Trust of the government (TOG) is citizens' belief in the ability and integrity of the government. Research studies on organisational behaviour showed that employees' trust in an organisation enables them to have positive attitudes and take positive actions. Regarding e-government adoption, numerous studies have proven that trust of the government has a positive influence on the citizens' intention to use e-government, which is an important predictor of citizens' e-government adoption [1,8,12–16]. When citizens have a higher level of trust in government, they tend to be close to government agencies (participate in public affairs, collect government information, and care about national affairs, etc.). Therefore, the first hypothesis of this study is proposed.

**Hypothesis 1 (H1).** *TOG positively influences citizens' e-government adoption*.

Trust of the internet (TOI), as institution-based trust, refers to one's perceptions of internet security policies and regulations that make him or her feel safe. As the internet is the carrier for delivering e-government services to citizens, trust of the internet is consistently identified as a key predictor of e-government adoption. According to existing studies, trust of the internet has a positive effect on e-government adoption [1,9,17]. These studies argued that citizens believe that the internet is reliable and secure, and could support error-free, secure transactions, and increase the users' willingness to use e-government. Influenced by the above literature, the second hypothesis of this study is proposed.

**Hypothesis 2 (H2).** *TOI positively influences citizens' e-government adoption*.

Perceived risk (PR), as citizens' subjective expectation of suffering a loss in pursuit of the desired outcome [18], is an important predictor of citizens' e-government adoption [4,6,15,19,20], as well as a crucial factor in m-commerce that influences user's use behaviour [21]. Regarding e-government, perceived risk is composed of the perception of exposure of personal information, privacy, and the loss of money. Numerous studies have already investigated perceived risk and its influence on citizens' e-government adoption, which found there is a significant and negative relationship between perceived risk and citizens' e-government adoption. The perceived risk reduces citizens' intentions to exchange information and transact affairs on the internet [18]. Drawing from these studies, the following hypothesis is proposed.

**Hypothesis 3 (H3).** *PR negatively influences citizens' e-government adoption*.

The relationship between trust and perceived risk has been explored by some empirical studies [1,8,9,22,23]. The results of these studies are different. For instance, Carter et al. 's study suggested that trust of the internet has a positive and significant effect on the perception of risk, and trust in government has a positive but not significant effect on perceived risk [9]. Lee and Song's study showed that trust in the capabilities of an

organisation that provides e-government services to citizens has a significant and negative effect on perceived risk [8]. However, these studies came from different countries, and thus, cultural factors might have influenced the results. Theoretically, trust can reduce uncertainty and risk. Therefore, this study proposes the following hypotheses.

**Hypothesis 4 (H4).** *Higher levels of TOG will reduce the PR of citizens' e-government adoption.*

**Hypothesis 5 (H5).** *Higher levels of TOI will reduce the PR of citizens' e-government adoption.*

In addition, this study tries to explore the moderating effect of perceived risk between trust and e-government adoption. A higher level of trust will increase the citizens' willingness to use e-government. However, the perception of risk will reduce citizens' attention to use e-government. Therefore, we want to know whether perceived risk plays a moderating role between trust and e-government adoption. From the citizens' perspective, if citizens have the same level of trust in government or the internet, they might tend to use e-government services. However, if they perceive high risks during this process, they might give up using e-government services. The perceived risk might have a negative moderating effect on the relationship between trust and e-government adoption. Depending on these arguments, this study proposes the following hypotheses.

**Hypothesis 6 (H6).** *PR has a negative moderating effect on the relationship between TOG and citizens' e-government adoption.*

**Hypothesis 7 (H7).** *PR has a negative moderating effect on the relationship between TOI and citizens' e-government adoption.*

2.2.2. UTAUT and Citizens' E-Government Adoption

UTAUT, as the classic information technology and information system acceptance model, is widely used in the field of information technology adoption and acceptance. UTAUT integrated eight models, such as the technology acceptant model (TAM), theory of reasoned action (TRA), theory of planned behaviour (TPB), motivation model (MM), innovation diffusion theory (IDT), and social cognition theory (SCT). UTAUT and extending the unified theory of acceptance and use of technology (UTATU2) proposed that performance expectancy (PE), effort expectancy (EE), social influence (SI), and facilitation conditions (FC) have a positive effect on the adoption of information technology or information systems [24,25]. These four major constructs: PE, EE, SI, and FC, present core constructs of other information adoption models such as perceived usefulness (PU), perceived ease of use (PEOU), and subjective norms (SN), etc. However, UTAUT was developed in the job context. Therefore, it is very necessary to verify the applicability of this model in e-government adoption. Fortunately, some existing studies have already verified that PE, EE, SI, and FC positively correlated with e-government adoption [26,27]. These studies argued that citizens' perception of usefulness, ease of use, social influence, convenient internet, and PC/mobile phone are significant to citizens' e-government adoption. Therefore, the following four hypotheses are offered.

**Hypothesis 8 (H8).** *EE will positively influence citizens' e-government adoption.*

**Hypothesis 9 (H9).** *PE will positively influence citizens' e-government adoption.*

**Hypothesis 10 (H10).** *SI will positively influence citizens' e-government adoption.*

**Hypothesis 11 (H11).** *FC will positively influence citizens' e-government adoption.*

In addition, there is also a relationship between PE and EE. In the TAM, PEOU has a significant positive effect on PU [28]. In the UTAUT, EE has a positive effect on PE [24,25].

Some studies have already used TAM and UTAUT in the area of e-government adoption, which also proved the positive and significant effect of PEOU on PU and EE on PE [8,29]. According to TAM and UTAUT, the hypothesis is proposed.

**Hypothesis 12 (H12).** *EE will positively influence PE.*

### 2.2.3. Trust and UTAUT

As aforementioned, the existing research has introduced trust into UTAUT to explore the influence of trust on the core variables of this model. Particularly in the field of e-business, existing research has also proven that trust has a significant impact on PE and EE [30–32]. Similarly, trust, as an external variable, has been introduced in TAM, and their results indicated that trust is positively associated with PU and PEOU [33]. In the research of e-government field, previous studies have drawn the similar conclusions. Lee and Song (2013) introduced the citizens' organisational trust into UTAUT, whose results also proved that trust has a positive effect on both PE and EE [8]. Although the concept of trust in these studies is multi-dimensional, it essentially covers the trust in organisations that provide services, and the internet or technology that deliver the services. Therefore, this study proposes the following four hypotheses.

**Hypothesis 13 (H13).** *TOG positively influences EE.*

**Hypothesis 14 (H14).** *TOI positively influences EE.*

**Hypothesis 15 (H15).** *TOG positively influences PE.*

**Hypothesis 16 (H16).** *TOI positively influences PE.*

### 2.2.4. Research Model

Based on the aforementioned literature, a model for analysing the role of trust and perceived risk in e-government adoption is proposed (Figure 1).

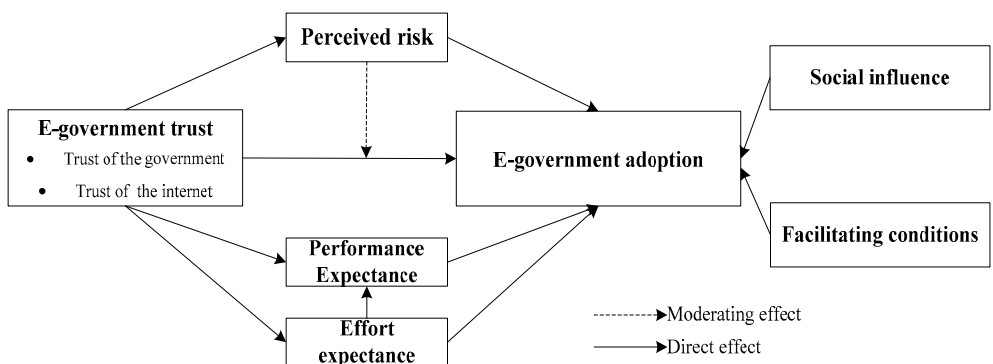

**Figure 1.** Research Model.

### 2.3. Methods

Studies of technology adoption have traditionally been conducted using survey research [1]. Consequently, this study surveyed a diverse group of citizens to obtain their perceptions of e-government services. The results were mainly analysed using a structural equation model (SEM) and multiple linear regression analysis. An important advantage of structural equation models (SEM) is their capacity to combine empirical observations with relations among unobserved constructs into a single integrated system [34]. Additionally, SEM is used to analyze the structural relationship between measured variables and latent constructs [35], which has been widely employed in the field of e-government adoption. This model was chosen as the aim of this study is to analyse the relationship between latent

variables which influence citizens' e-government adoption. Multiple linear regression analysis is used to test the moderation effect of PR on the relationship between trust and EGA, including three steps: this study mean centered both predictors first; then multiplied the centered predictors into an interaction predictor variable; finally, we entered both mean centered predictors and the interaction predictor into a regression analysis [36]. Both the SPSS 21 and Amos 21 software were used during this process.

## 3. Survey Design

### 3.1. Measurement

To test the research model proposed for this study, a questionnaire survey was used. Survey items were derived from previous studies. There are eight variables in the model. TOG was measured by using three items derived from these studies [9,26] and has been slightly modified to fit this study. TOI was measured with two items that were borrowed and modified from the existing literature [9,26,27]. PR was supported by two items, which were borrowed and modified from the existing literature [7,9,23,37–39]. EE, PE, SI, and FC were derived from the UTAUT model [24]. EE was measured with four items, PE was measured with three items, SI was measured with three items, and FC was measured with three items, which were borrowed and modified from the existing literature [1,23,26,27]. E-government adoption (EGA) was supported by three items [21,40] and represents citizens' intention to inquire for information, complete transactions, and share opinions through e-government websites. Table 1 shows all constructs and items.

**Table 1.** Constructs and items.

| Constructs | | Items |
|---|---|---|
| Trust of the government (TOG) | 1. | I trust the government. |
| | 2. | I think the government is trustworthy. |
| | 3. | I feel that most things the government does are correct. |
| Trust of the internet (TOI) | 1. | Overall, the Internet environment is safe. |
| | 2. | In the process of using the Internet, laws, and policies can protect me from various unsafe factors. |
| Perceived risk (PR) | 1. | I feel that there is no risk in using government websites(reverse question). |
| | 2. | In general, I feel that government website services are safe and reliable(reverse question). |
| Effort expectancy (EE) | 1. | I have knowledge and skills to use government websites. |
| | 2. | For me, learning to use government websites is very easy. |
| | 3. | Even if I have never used a similar network service, I am confident in using government website services. |
| | 4. | Even without the help of others, I can learn to use government websites. |
| Performance Expectancy (PE) | 1. | The government websites provided me with useful information. |
| | 2. | The government websites provided me with valuable services. |
| | 3. | The government websites provide me with channels for expressing opinions and suggestions. |
| Social Influence (SI) | 1. | People I know think I should use government websites. |
| | 2. | The media believe that the public should use government websites. |
| | 3. | As far as I know, everyone around me is using e-government services. |
| Facilitating Conditions (FC) | 1. | I often use the computer or smartphone. |
| | 2. | I can get a high-speed internet connection at home. |
| | 3. | I can get a high-speed internet connection in the workplace. |
| E-government adoption (EGA) | 1. | I am willing to inquiry about information from e-government websites. |
| | 2. | I am willing to use online services such as online tax payment, online approval, and online certificate application. |
| | 3. | I am willing to share opinions with e-government websites. |

In addition, demographic characteristics also impact citizens' e-government adoption. Therefore, several demographic variables were also collected. The questionnaire was divided into two sections: the first section is the normal multiple-choice question for the personal information; the second section is the 5-point Likert scale questions (i.e., Strongly Disagree, Disagree, Neutral, Agree, and Strongly Agree). As English is not the first language in China, the questionnaire was prepared in Chinese. Back translation was used, with the questionnaire translated from English to Chinese first, and then from Chinese to English. A pre-test was done using 10 doctoral students and 50 citizens to improve the quality of survey items. The final questionnaire includes the eight scales, as well as demographics.

### 3.2. Data Collection

To collect data, a paper-based survey was used. Questionnaires were distributed to 1500 citizens within the cities of Chengdu, Changsha, Pudong, Shenyang, and Shenzhen from July to August 2018. These cities are distributed in eastern, central, and western China, which can represent the status quo of Chinese government websites. A convenient sampling method was adopted. Researchers distributed questionnaires in some places with dense populations, such as parks and libraries. To ensure that the interviewees fill in the questionnaire carefully, we prepared gifts for them as an incentive. There were 1251 questionnaires returned, but some respondents were under the age of 18, and some respondents did not fill in the questionnaires carefully, which were eliminated. Thus, we have 966 effective samples to verify our model. Among them, 51% were males, 49% were females. Many respondents (76%) were in the age group from 20–49. Most respondents (63.1%) had a college degree. The majority of respondents had more than 6 years of internet experience (78.9%). In summary, the respondents were highly educated, mature adults familiar with the internet. Table 2 shows the demographic information of samples.

**Table 2.** Demographic Statistics (*n* = 966).

|  | Frequency | Percentage (%) |  | Frequency | Percentage (%) |
|---|---|---|---|---|---|
| Gender |  |  | Internet experience |  |  |
| male | 493 | 51.0 | <1 year | 29 | 3.0 |
| female | 473 | 49.0 | 1–5 years | 174 | 18.0 |
|  |  |  | 6–10 years | 355 | 36.7 |
| Age (years) |  |  | >10 years | 408 | 42.2 |
| 18–20 | 28 | 2.9 |  |  |  |
| 20–29 | 422 | 43.7 | Education |  |  |
| 30–39 | 312 | 32.3 | Under- high school | 71 | 7.3 |
| 40–49 | 116 | 12.0 | High school | 126 | 13.0 |
| 50–59 | 61 | 6.3 | Graduate | 609 | 63.1 |
| >59 | 27 | 2.8 | Post-graduate | 160 | 16.6 |

## 4. Data Analysis and Results

Three stages data analysis were adopted in the study. The first stage is descriptive statistics, including the means and standard deviations of the variables, and correlation coefficients. The second stage is structural equation model, including measurement model and structural model. The third stage is analysing the moderating effect of perceived risk on the relationship between trust and citizens' EGA.

### 4.1. Descriptive Statistics and Correlation

Means and standard deviations of TOG, TOI, PR, EE, PE, SI, FC, and EGA are shown in Table 3, which indicate that the respondents have a high level of trust, low perception of risk, and strong willingness to use e-government websites. The standard deviations ($\leq 1$) further indicate that the respondents' views on the sub-items of these variables are relatively consistent, and their opinions are relatively uniform.

**Table 3.** Cronbach's alpha value and Pearson correlation.

|  | TOG | TOI | PR | EE | PE | SI | FC | EGA |
|---|---|---|---|---|---|---|---|---|
| TOG | 1 |  |  |  |  |  |  |  |
| TOI | 0.465 ** | 1 |  |  |  |  |  |  |
| PR | −0.435 ** | −0.523 ** | 1 |  |  |  |  |  |
| EE | 0.287 ** | 0.301 ** | −0.183 ** | 1 |  |  |  |  |
| PE | 0.512 ** | 0.432 ** | −0.449 ** | 0.270 ** | 1 |  |  |  |
| SI | 0.437 ** | 0.360 ** | −0.316 ** | 0.334 ** | 0.510 ** | 1 |  |  |
| FC | 0.175 ** | 0.170 ** | −0.140 ** | 0.375 ** | 0.203 ** | 0.241 ** | 1 |  |
| EGA | 0.312 ** | 0.334 ** | −0.209 ** | 0.494 ** | 0.405 ** | 0.348 ** | 0.437 ** | 1 |
| Mean | 3.615 | 3.181 | 2.794 | 3.722 | 3.578 | 3.339 | 3.908 | 3.948 |
| S.D. | 0.916 | 0.849 | 1.000 | 0.850 | 0.821 | 0.824 | 0.824 | 0.796 |

** Correlation is significant at the 0.01 level (two-tailed).

The correlation coefficient for each pair of factors is also depicted in Table 3. All correlation coefficients are positively correlated with each other, except PR. PR is negatively correlated with EE, PE, SI, FC, and EGA. These values further indicate that all variables are correlated, and can be analysed with SEM and regression.

Similarly, the study also examines common method bias (CMB), as the data of all the constructs (a questionnaire) originated from the same respondents. In this study, some procedural, as well as statistical remedies were applied. For procedural CMB issues, valid scales, lucid language, etc., measures were applied. For statistical remedies, Herman's single factor scores were tested. The total variance for a single factor is 32.745%, which is lower than the suggested threshold of less than 50% [41]. Hence, this result indicates the data has no issues of CMB.

### 4.2. Structural Equation Model

This study uses a structural equation model (SEM) with AMOS 21 to test the causal relationships of the model and the hypotheses of this study. As we measured all variables using a 5-point Likert scale, data gathered from the survey is continuous. Therefore, the SEM with maximum likelihood (ML) estimation is appropriate for this type of data. First, this study uses confirmation factor analysis (CFA) to analyse the reliability and validity of this model. Second, this study employs a structural model to analyse the impact of TOG, TOI, PR, EE, PE, SI, and FC on EGA. Similarly, this study chooses Chi-square, adjusted for the degree of freedom (CMIN/DF), Goodness of Fit Index (GFI), Comparative Fit Index (CFI), Incremental Fit Index (IFI), and Root Mean Square Error of Approximation (RMSEA) as model fit indices based on previous studies [42].

#### 4.2.1. Measurement Model

The results of CFA show the values of fit indicators: CMIN/DF = 2.593, GFI = 0.957, CFI = 0.974, IFI = 0.974, and RMSEA = 0.061, exhibit good model fit. Composite reliability (CR), average variance extracted (AVE), and Cronbach's $\alpha$ value of all variables and factor loadings were also calculated. The general indicator for reliability concerning factor loadings is that items should be above 0.7, although values above 0.5 are acceptable in the SEM [43]. These are achieved, as can be seen from Table 4, where factor loadings range from 0.523 to 0.947. Construct reliability was evaluated by using Cronbach's $\alpha$ and composite reliability. Cronbach's $\alpha$ value needed to be at least 0.7 [44]. This was achieved. Constructs' Cronbach's $\alpha$ values range from 0.699 to 0.924. Similarly, the criterion for composite reliability is based on the view that appropriate values should be above 0.7, although values above 0.6 are acceptable [43]. All the composite reliability values displayed in Table 4 meet the 0.6 criteria, as the values range from 0.677 to 0.894. As such, both the Cronbach's $\alpha$ and composite reliability criteria are good, confirming the construct reliability of the factors. Additionally, the AVE was used to evaluate convergent validity based on the criteria that valid constructs should have AVE values above 0.5 [43]. These criteria are also meet, as AVE values range from 0.514 to 0.744.

**Table 4.** Results of reliability and validity test.

|  |  | Cronbach's $\alpha$ | Factor Loadings | CR | AVE |
|---|---|---|---|---|---|
| TOG | TOG1<br>TOG2<br>TOG3 | 0.924 | 0.879<br>0.947<br>0.903 | 0.894 | 0.744 |
| TOI | TOI1<br>TOI2 | 0.837 | 0.653<br>0.775 | 0.677 | 0.514 |
| PR | PR1<br>PR2 | 0.858 | 0.853<br>0.881 | 0.858 | 0.752 |
| EE | EE1<br>EE2<br>EE3<br>EE4 | 0.866 | 0.717<br>0.807<br>0.830<br>0.801 | 0.869 | 0.624 |
| PE | PE1<br>PE2<br>PE3 | 0.830 | 0.860<br>0.874<br>0.744 | 0.878 | 0.706 |
| SI | SI1<br>SI2<br>SI3 | 0.753 | 0.672<br>0.798<br>0.674 | 0.843 | 0.643 |
| FC | FC1<br>FC2<br>FC3 | 0.699 | 0.706<br>0.523<br>0.768 | 0.823 | 0.614 |
| EGA | EGA1<br>EGA2<br>EGA3 | 0.833 | 0.753<br>0.816<br>0.804 | 0.868 | 0.687 |

### 4.2.2. Structural Model Testing

This study uses AMOS 21 to analyse the model of this study. The values of fit indicators, CMIN/DF = 4.636, GFI = 0.92, CFI = 0.933, IFI = 0.933, and RMSEA = 0.061, exhibit good model fit.

The results of SEM are shown in Table 5. Table 5 shows the path coefficients, critical ratios (C.R), probability values (P), and hypotheses testing results. As Table 5 shows, the path coefficients between TOG and EGA are not statistically significant; H1 is not supported by data, while TOI has a positive and significant effect on citizens' government adoption; H2 is supported. PR is positively and significantly associated with citizens' government adoption. The result does not support H3. Similarly, TOG and TOI are negatively and significantly associated with PR. Hence, H4 and H5 are supported by data as expected. The negative relations also appear to be quite reasonable, as citizens' higher governmental and internet trust levels can reduce the perceived risk of using e-government websites.

Likewise, the results of path analysis also support H8, H9, H10, and H11. That means citizens' PE, EE, SI, and FC are positively and significantly related to e-government adoption. In addition, EE can positively and significantly influence PE, therefore H12 is supported. These results are reasonable as citizens' performance expectancy and effort expectancy increase the citizens' intentions to use e-government. Social influence and facilitating conditions also positively influence the citizens' intentions to use e-government. Citizens' effort expectance increases their performance expectance. Similarly, the relationship between TOG and EE, TOI and EE, TOG and PE, and TOI and PE are positive and significant. H13, H14, H15, and H16 are supported. These results mean trust plays a vital role in the UTAUT, which influence the core variables PE, EE, and PR.

**Table 5.** Path coefficients of structural equation model.

|  |  | Path Coefficient | C. R | *p*-Value | Result Supported/Rejected |
|---|---|---|---|---|---|
| H1 | TOG→EGA | −0.018 | −0.444 | 0.657 | rejected |
| H2 | TOI→EGA | 0.239 ** | 3.243 | 0.001 | supported |
| H3 | PR→EGA | 0.127 | 2.377 | 0.017 | rejected |
| H4 | TOG→PR | −0.107 ** | −2.407 | 0.016 | supported |
| H5 | TOI→PR | −0.638 *** | −10.999 | 0.000 | supported |
| H8 | EE→EGA | 0.272 *** | 6.453 | 0.000 | supported |
| H9 | PE→EGA | 0.256 *** | 5.577 | 0.000 | supported |
| H10 | SI→EGA | 0.069 ** | 2.180 | 0.029 | supported |
| H11 | FC→EGA | 0.378 *** | 7.975 | 0.000 | supported |
| H12 | EE→PE | 0.108 ** | 3.214 | 0.001 | supported |
| H13 | TOG→EE | 0.122 ** | 2.766 | 0.006 | supported |
| H14 | TOI→EE | 0.244 *** | 4.850 | 0.000 | supported |
| H15 | TOG→PE | 0.320 *** | 7.551 | 0.000 | supported |
| H16 | TOI→PE | 0.367 *** | 7.284 | 0.000 | supported |
| Added | FC→PE | 0.432 *** | 9.103 | 0.000 | supported |

Note: ** $p < 0.01$, and *** $p < 0.001$.

This study further analysed the path relationship between FC and EE while running this model, as shown in Table 5, which is also positive and significant. This result is reasonable according to previous studies [37].

In addition, the coefficients of determination, which are also denoted as R-Squared ($R^2$), were estimated as an essential criterion for assessing the endogenous latent variables of the structural model. $R^2$ refers to the proportion of the variance of the dependent variable that is predictable from the independent variable [42]. $R^2$ values range between 0 and 1. In Table 6, an $R^2$ value of 0.296 means that the predictors of EE (FC) explain about 29.6% of its variance, 0.498 means that the predictors of PR (TOG and TOI) explain about 49.8% of its variance, 0.428 means that the predictors of PE (TOG, TOI, and EE) explain about 42.8% of its variance, and 0.516 means that the predictors of EGA (TOG, TOI, PR, EE, PE, FC, and SI) explain about 51.6% of its variance, the residual being due to the error variance, etc.

**Table 6.** Coefficients of determination.

| Constructs | $R^2$ |
|---|---|
| EE | 0.296 |
| PR | 0.498 |
| PE | 0.428 |
| EGA | 0.516 |

Moreover, Tables 7–9 show the standardized direct, indirect, and total effects of the structural model, respectively. As an example, the direct effect of TOI on EGA is 0.239, and its indirect effect is 0.086; therefore, the total effect of PU on EGA is 0.325 = 0.239 + 0.086, which means that, when TOI increases by 1 standard deviation, EGA increases by 0.325 of a standard deviation.

**Table 7.** Direct effect in the structural model.

|  | TOI | SI | FC | TOG | EE | PR | PE |
|---|---|---|---|---|---|---|---|
| EE | 0.244 |  | 0.432 | 0.122 |  |  |  |
| PR | −0.638 |  |  | −0.107 |  |  |  |
| PE | 0.367 |  |  | 0.320 | 0.108 |  |  |
| EGA | 0.239 | 0.069 | 0.378 | −0.018 | 0.272 | 0.127 | 0.256 |

**Table 8.** Indirect effect in the structural model.

|  | TOI | SI | FC | TOG | EE | PR | PE |
|---|---|---|---|---|---|---|---|
| EE | | | | | | | |
| PR | | | | | | | |
| PE | 0.026 | | 0.047 | 0.013 | | | |
| EGA | 0.086 | | 0.129 | 0.105 | 0.028 | | |

**Table 9.** Total effect in the structural model.

|  | TOI | SI | FC | TOG | EE | PR | PE |
|---|---|---|---|---|---|---|---|
| EE | 0.244 | | 0.432 | 0.122 | | | |
| PR | −0.638 | | | −0.107 | | | |
| PE | 0.393 | | 0.047 | 0.333 | 0.108 | | |
| EGA | 0.325 | 0.069 | 0.507 | 0.087 | 0.299 | 0.127 | 0.256 |

4.2.3. Moderation Analysis

Moderation analysis consists of five multiple regression models. In the multi-regression model, e-government adoption is the dependent variable. Independent variables contain PR, TOG, TOI, and the interaction term (PR×TOG or PR×TOI). As citizens' demographic characters have an impact on the adoption of e-government services, this study takes gender, age, education, and internet experience as control variables.

The results are displayed in Table 10, including unstandardized regression coefficients and standard errors. Unstandardized regression coefficients indicate the individual effect of X (independent variables) on Y (dependent variable). For instance, in model 1, the regression coefficient 0.101 indicates a change of 1 unit in the gender is associated with a change of 0.101 units in the outcome EGA. As shown in Table 10, model 1, model 2, and model 3 are used to test the moderating effect of PR on the relationship between TOG and e-government adoption. The regression coefficient of the interaction term (PR×TOG) is significant ($p < 0.05$), which means that there is a moderating effect of perceived risk on the relationship between TOG and e-government adoption. H6 is supported by the data.

**Table 10.** The effect of TOG and TOI on EGA is moderated by perceived risk.

|  | Model 1 | Model 2 | Model 3 | Model 4 | Model 5 |
|---|---|---|---|---|---|
| Gender | 0.101 * | 0.094 * | 0.093 * | 0.111 * | 0.109 * |
|  | (0.05) | (0.047) | (0.047) | (0.046) | (0.046) |
| Age | −0.018 | −0.051 * | −0.054 * | −0.041 * | −0.041 |
|  | (0.024) | (0.023) | (0.022) | (0.022) | (0.022) |
| Education | 0.029 | 0.033 | 0.034 | 0.044 * | 0.046 * |
|  | (0.024) | (0.023) | (0.023) | (0.023) | (0.023) |
| Internet experience | 0.214 *** | 0.222 *** | 0.219 *** | 0.216 *** | 0.217 *** |
|  | (0.033) | (0.031) | (0.031) | (0.031) | (0.031) |
| PR | | −0.098 *** | −0.094 *** | −0.065 * | −0.067 * |
|  | | (0.026) | (0.026) | (0.027) | (0.027) |
| TOG | | 0.236 *** | 0.24684 *** | | |
|  | | (0.028) | (0.029) | | |
| PR×TOG | | | −0.054 * | | |
|  | | | (0.022) | | |
| TOI | | | | 0.289 *** | 0.289 *** |
|  | | | | (0.032) | (0.032) |
| PR×TOI | | | | | −0.039 * |
|  | | | | | (0.019) |
| $R^2$ | 0.066 | 0.182 | 0.188 | 0.192 | 0.195 |
| Adjusted $R^2$ | 0.062 | 0.177 | 0.182 | 0.187 | 0.190 |
| $\Delta R^2$ | 0.066 *** | 0.117 *** | 0.005 * | 0.126 *** | 0.03 * |

Note: * $p < 0.05$, *** $p < 0.001$.

Similarly, model 1, model 4, and model 5 are used to test the moderating effect of PR on the relationship between TOI and e-government adoption. As Table 10 shows, the

regression coefficient of the interaction term (PR×TOI) is significant ($p < 0.05$), which means that there is a moderating effect of PR on the relationship between TOI and e-government adoption. H7 is supported by the data. This further signifies that the PR weakens the predictive power of TOG and TOI in determining the citizens' adoption of e-government services.

The graphic representations of perceived risk moderating the relationships between TOG and e-government adoption, as well as TOI and e-government adoption, are depicted in Figures 2 and 3. As Figures 2 and 3 show, the relationships between TOG and EGA, and TOI and EGA are different between individuals with high-risk perception and low-risk perception. The risk perception from low to high weakens the effect of TOG and TOI on citizens' e-government adoption, which denotes there is the simultaneous effect of trust and perceived risk on citizens' e-government adoption. Perception of risk is an important predictor of citizens' e-government adoption, which negatively moderates the relationship between trust and e-government adoption.

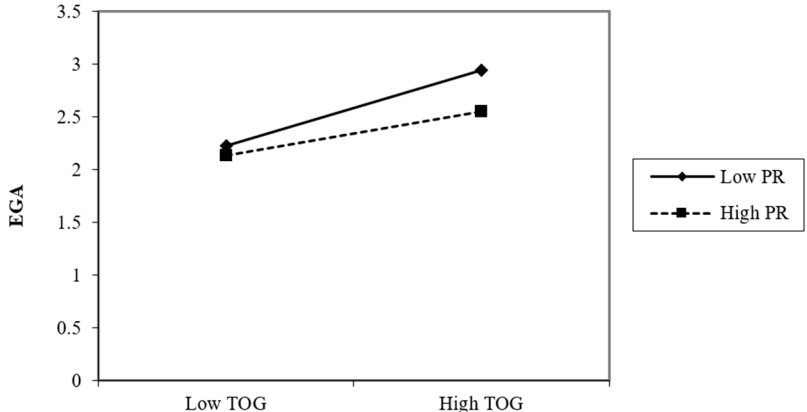

**Figure 2.** The plot of the moderating effect of perceived risk on the relationship between TOG and EGA.

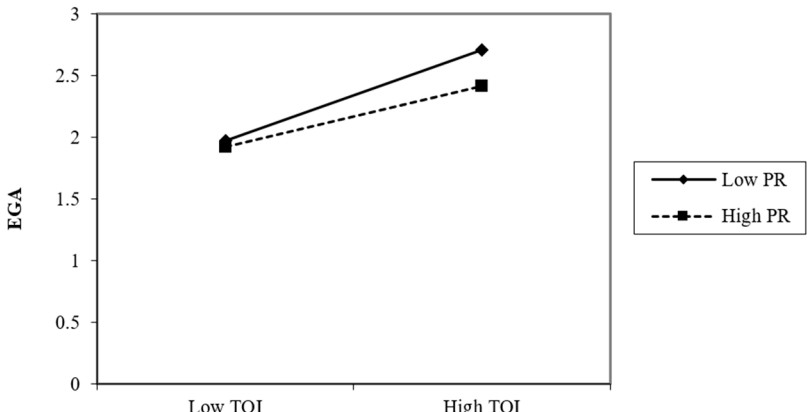

**Figure 3.** The plot of the moderating effect of perceived risk on the relationship between TOI and EGA.

Simultaneously, the regression analysis also indicates that TOG has a significant and positive impact on citizens' e-government adoption, and the perception of risk has a significant and negative effect on citizens' e-government adoption. H1 and H3 are supported by the regression analysis. The results of the regression analysis are inconsistent with those of the structural equation model. The possible reason is that the relationships between the variables in the structural equation model are more complicated, and they influence one another, which may cause the results to be statistically insignificant or different, namely, differences in analysis methods lead to differences in results. According

to the previous studies and the case of e-government, this study accepts the results of regression analysis, and believes that government trust increases users' willingness to use e-government, and the perceived risk reduces users' willingness to use e-government.

In addition, this study finds some significant conclusions regarding effect of demographic variables on users' behaviour. Compared with females, males are more willing to use e-government services. The influence of education level and age are unstable when adding other variables. Internet experience has a positive and significant impact on users' behaviour, namely users who have used the internet for a long time are more willing to use e-government services, which is consistent with previous studies suggesting that computer efficacy has a positive impact on users' behaviour.

## 5. Discussion and Conclusions

### 5.1. Discussion

This study develops an integrated model to analyse how trust and risk influence citizens' e-government adoption. Structural equation models and multiple linear regression analysis were employed to test these hypotheses. The results indicate that trust (TOI and TOG), PR, EE, PE, SI, and FC are significant predictors for the adoption of e-government services; trust reduces citizens' risk perception, and increases citizens' perception of benefits and ease of use for e-government services. Simultaneously, the moderation analysis shows that the moderating effect of perceived risk between TOG and the citizens' adoption of e-government, and TOI and the citizens' adoption of e-government services, are significant.

TOG and TOI are important predictors to explain citizens' e-government adoption. This study shows that the relationships between TOG/TOI and e-government adoption are positive and significant, which are consistent with previous studies [1,9,14]. PR is also a significant predictor of e-government adoption, which has a negative relationship to e-government adoption [8,18]. This study confirmed this conclusion. When citizens are aware of the risks when inquiring about information or using online services, they might go to brick-and-mortar government departments, or contact some acquaintance to collect information or apply for services. As a result, perceived risk reduces the willingness of using e-government. This argument justifies the finding of statistical analysis, which depicts negative cause-and-effect relations between PR and e-government adoption.

The relationship between trust and perceived risk has been tested by several previous studies. Their results indicated that trust has a negative relationship with perceived risk [1,9,14]. The results of this study are consistent with previous studies. When citizens have a high level of trust in government, they might tend to think the service provided by the government is reliable and risk-free. When users have a high level of trust in the internet, they might tend to believe that they use e-government websites safely. Simultaneously, the perception of risk can weaken the effect of trust on EGA.

This study also finds that other variables: PE, EE, SI, and FC, are important for citizens to adopt e-government. These findings are supported by UTAUT and UTAUT2 [24,25]. Based on UTAUT, constructs PE, EE, and SI influence the acceptance and use of information technology. Moreover, UTAUT2 pointed out that FC has a positive effect on the acceptance and use of information technology. From the literature review, we have found that PE, EE, SI, and FC have a positive relationship with the use of e-government [8,45]. Additionally, one recent study also confirmed that PE, EE, SI, and FC have a positive influence on users' intention to use working from home technologies during the COVID-19 pandemic [46]. If citizens can use e-government and perceive that this website is useful, they will most likely adopt it. After that, citizen's e-government adoption will be influenced by friends and family and public media; if their friends and family are using e-government websites, and the media are advertising the benefits of using the e-government websites, they will be more likely to use them. Simultaneously, when they have a personal computer, laptop, or smartphone and internet access, they will be more willing to use e-government websites.

Based on data analysis, this study finds that FC has a positive and significant effect on EE, although UTAUT has not tested the relationship between FC and EE. This result

is reasonable and similar to other studies [47]. If users have more devices to connect to e-government websites, they tend to perceive that they have more ability to use e-government websites.

From the structural equation model analysis, we can see that trust of the government and trust of the internet have a positive effect on PE and EE. Research from Korea has already found that the trust of public service providers can improve citizens' PE and EE [8]. When users have a high level of government and internet trust, they tend to make a good impression of the services delivered by the government. Hence, they will think e-government websites are more useful, and they can use them.

### 5.2. Theoretical Implications

Theoretically, this study has three contributions. First, this study expanded the UTAUT model. The UTAUT model was developed by Venkatesh in 2003 [24], which showed that EE, PE, and SI can influence intention to use information technology, and FC can influence actual user behaviour. Venkatesh revised the UTAUT and proposed UTAUT2 in 2012 [25]. UTAUT2 pointed out that EE, PE, SI, and FC can influence user's intention to use information technology. This model was developed in the job environment. Therefore, previous studies always tried to use UTAUT in the e-government field by adding some new constructs, to predict the user's e-government adoption. This study also proved that the UTAUT model is applicable in the area of e-government adoption. Simultaneously, this study proposed a model composed of trust, perceived risk, and the UTAUT model, and further tested the impact the government trust and internet trust on core variables (effort expectancy and performance expectancy) of the UTAUT model, and on citizens' e-government adoption. This study also confirmed the influence of risk perception on citizens' e-government adoption.

Second, this study updated the UTAUT model, which found FC affects effort expectancy (EE), and the relationship between FC and EE is positive. Research from India has also drawn the same conclusion [47]. When users are in the context of e-government services, they have more convenient devices to connect to e-government websites, they will have a stronger ability to use government websites.

Third, the model of this study developed include trust and risk perception and tested the relationship between them. This study further confirmed the moderating effect of perceived risk between trust and e-government adoption. Compared to previous research [7], this study analysed the relationship between trust and risk perception in more depth. In theory, it helps us clarify the relationship between trust and risk perception, and in practice, it helps us understand users' e-government adoption.

### 5.3. Pratical Implications

In practice, based on the findings of this study, we recommend some essential points to public managers of Chinese e-government, as well as other governments and policy-makers.

First, trust of the government and the internet are vital issues to improve e-government adoption. These factors affect perceived risk, perception of usefulness, whether e-government is easy to use, and e-government adoption. The government should continuously improve the credibility of the government and protect internet security. In addition, facing the digital transformation of the whole society, the government should introduce some plans or projects to improve the digital skills of the public. For example, scholars have proposed that, in the context of Industry 4.0, small and medium-sized enterprises (SMEs) need to improve the skills of their employees for adapting to the development of technology [48].

Second, government agencies should focus on how they can ensure citizens have enough devices to use e-government. Facilitating conditions is an important aspect of encouraging users to use the e-government system. In addition, with the application of multiple emerging technologies in e-government in China, such as artificial intelligence and blockchain, how to use e-government services is getting more and more complicated

for elderly adults who are incapable of using a personal computer or mobile phone. We should provide facilitating conditions or a variety of offline services for them. For instance, the issue of digital exclusion in an aging society has already been discussed, which provides some directions for us [49].

Third, the citizens' e-government adoption can be influenced by friends, family, and public media. Hence, the government should make full use of the internet, public knowledge bulletin boards, newspapers, television, radio, and social media to make citizens aware of and familiar with the list of online services.

### 5.4. Limitations and Future Studies

There are few limitations of this study outlined within this section. First, the research object of this study is only e-government websites. Although e-government websites are the most important channel to deliver online services to citizens, exploring citizens' adoption of e-government websites can help academics, practitioners, researchers, and policy-makers understand citizens' use behaviour of other channels, such as Weibo, APP, etc. However, future studies are necessary to research citizens' adoption of other channels to plot the panorama of citizens' use of e-government in China. Second, in the interest of parsimony and obtaining as many participants as possible, the model of this study is tested by Chinese citizen user samples. Future studies should expand samples, especially cross-region (urban areas and rural areas), cross-culture, or cross-country samples, to present a more comprehensive view of e-government adoption. Third, this study only focused on the users' adoption of G2C. Recently, various digital technologies have been widely used in the government regulation of the industry, such as the milk-producing industry [50]. Therefore, future research needs to pay more attention to the government's adoption of information technology and its influencing factors.

**Funding:** This research was funded by the youth project of Humanities and Social Science of Ministry of Education, China, Grant Number 20YJC630066, Tsinghua University Initiative Scientific Research Program, Grant Number 20205080071, and China postdoctoral science foundation, Grant Number 2019M660704.

**Institutional Review Board Statement:** Not applicable.

**Informed Consent Statement:** Not applicable.

**Data Availability Statement:** Data sharing not applicable.

**Conflicts of Interest:** The author declares no conflict of interest.

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
