# Peer review of "The Role of Trust and Risk in Citizens’ E-Government Services Adoption: A Perspective of the Extended UTAUT Model"

_sustainability, doi:10.3390/su13147671_

Round 1

Reviewer 1 Report

  • Survey sample seems to be from urban context and relatively young people that may not be representative of population.
  • 4.2 Title in capital letters.
  • The number of table 4 in text is not correct.
  • Range of composite reliability is not correct in the text.
  • 4.2.1 and 4.2.2 have the same title which can not be.
  • Check text before table 5. Sometimes it seems that one thing is true and after authors say that the results don´t support the Hypothesis.
  • Figures 2 and 3 are very similar and have the same caption.
  • Methods could be better explained in chapter 2 (Ex.: 2. Research context, Hypotheses and Methods)
  • References are not in the correct format.
  • Novelty of work developed is not clear.

Author Response

Dear professor,

Thank you very much for your suggestions and comments.

I have responded in detail to your comments one by one and modified the manuscript.  

Thanks again.

Best wishes to you.

Wenjuan

Reviewer 2 Report

The research presented is of high quality and well written. The description of the research path is objective and clear.

Despite this, the paper abstract could be improved. It is very technical and accurate, but it is not written most appealingly. After reading it, we understand the research and what it achieved. However, it does not make us want to read it. On the contrary, it is possible to think that after reading the abstract, we read it all. So maybe if the authors could improve the abstract, making it more appealing.

Another improvement issue is the way the sample was defined. It is understandable the convenience for the sample, but why parks and libraries? And why was the questionnaire distributed to young children (under 18?) They shouldn't have. Please explain better this.

Author Response

(The authors gave the same response as above.)

Reviewer 3 Report

I enjoyed your article very much. I found it clear, carefully reasoned and cogent. I suspect most of the issues I had were linked to English usage so i do encourage a close copy and line edit of your effort. That said, one does not "implement" but rather "employ" a theory (line 11). Similarly, you did not undertake a 996 survey but obtained 996 usable responses from your sample (Line 13). I was also unclear on what you meant in lines 39-40. Here are three or four other questions: when you use "moderating" do you mean mediating? What exactly have been China's "remarkable achievements in e-governance in recent years? Increased usage and availability and how have those changes improved services? I wondered about your construct: " I feel there is no risk in using government websites? " Presumptively, there is ALWAYS some measure of risk at play? Likewise your use of "feel" struck me. You cannot judge feelings only perceptions and should ask for the same?

Author Response

(The authors gave the same response as above.)

Round 2

Reviewer 1 Report

I have read the answer of authors to my review. From their answers it seems that all comments were addressed.